# The Influence of BMP6 on Serotonin and Glucose Metabolism

**DOI:** 10.3390/ijms25147842

**Published:** 2024-07-18

**Authors:** Marina Milešević, Ivona Matić Jelić, Viktorija Rumenović, Natalia Ivanjko, Slobodan Vukičević, Tatjana Bordukalo-Nikšić

**Affiliations:** Laboratory for Mineralized Tissues, Center for Translational and Clinical Research, School of Medicine, University of Zagreb, 10000 Zagreb, Croatia; marina.milesevic@mef.hr (M.M.); ivona.jelic@mef.hr (I.M.J.); viktorija.rumenovic@mef.hr (V.R.); natalia.ivanjko@mef.hr (N.I.); slobodan.vukicevic@mef.hr (S.V.)

**Keywords:** BMP6 knockout mice, serotonin, glucose metabolism, insulin, pancreas

## Abstract

Previous studies have suggested a potential role of bone morphogenetic protein 6 (BMP6) in glucose metabolism, which also seems to be regulated by serotonin (5-hydroxytryptamine, 5HT), a biogenic amine with multiple roles in the organism. In this study, we explored possible interactions between BMP6, serotonin, and glucose metabolism regulation. The effect of BMP6 or 5HT on pancreatic β-cells has been studied in vitro using the INS-1 832/13 rat insulinoma cell line. Studies in vivo have been performed on mice with the global deletion of the *Bmp6* gene (BMP6−/−) and included glucose and insulin tolerance tests, gene expression studies using RT-PCR, immunohistochemistry, and ELISA analyses. We have shown that BMP6 and 5HT treatments have the opposite effect on insulin secretion from INS-1 cells. The effect of BMP6 on the 5HT system in vivo depends on the tissue studied, with no observable systemic effect on peripheral 5HT metabolism. BMP6 deficiency does not cause diabetic changes, although a mild difference in insulin tolerance test between BMP6−/− and WT mice was observed. In conclusion, BMP6 does not directly influence glucose metabolism, but there is a possibility that its deletion causes slowly developing changes in glucose and serotonin metabolism, which would become more expressed with ageing.

## 1. Introduction

Osteoblasts, bone-forming cells primarily involved in osteogenesis, also have a significant role in whole-body energy homeostasis and secrete several hormones with endocrine effects. Among them, osteocalcin, a product of differentiated osteoblasts, is involved in the regulation of insulin secretion and glucose metabolism [1]. Besides osteocalcin, bone morphogenetic proteins (BMPs), members of the transforming growth factor β (TGFβ) superfamily, well-known for their signalling role in osteoblast and osteoclast cells [2], have recently been recognised as additional important factors in energy metabolism [3,4]. In particular, BMPs seem to play a significant role in glucose metabolism, and different BMP ligands, receptors, and downstream signalling molecules have been found in pancreatic islet cells [5]. It has been shown that BMP7 promotes the exocrine-to-endocrine conversion of pancreatic tissue in vitro and in vivo [6], whereas circulating BMP9 improves glucose homeostasis in newly diagnosed subjects with type 2 diabetes [7]. Further, BMP7 and BMP9 also contribute to gluconeogenesis and improve insulin sensitivity in the liver [8,9]. In our recently published study, the treatment of *ob*/*ob* mice with BMP6 significantly reduced circulating glucose and lipid concentrations while increasing plasma insulin levels, emphasising the potential role of BMP6 in the regulation of glucose metabolism [10].

Mice with global or conditional deletion of BMP ligands or their receptors are widely used animal models in the research of different aspects of BMP function. While some of the systemic BMP deletions are lethal [2], mice with globally deleted BMP6 demonstrated only minor skeletal defects [11]. In addition, our recent work revealed the changes in exocrine pancreatic tissue of BMP6 knockout mice, showing iron deposits and pancreatic degeneration [12].

Serotonin (5-hydroxytryptamine, 5HT) is mostly known as a neurotransmitter regulating complex behavioural patterns [13]. Outside the central nervous system, 5HT is involved in multiple cellular and metabolic functions, and its role in energy metabolism has been extensively studied [14]. In addition to its well-defined role in the regulation of gastrointestinal motility, 5HT has been recognised as a factor in the regulation of glucose homeostasis [15], as well as in bone metabolism, where elevated circulating 5HT level resulted in bone loss due to the increased bone turnover rate [16]. Although the vast majority of peripheral 5HT is synthesised within the enterochromaffin cells of the gut and stored in platelets, it is also produced by pancreatic β-cells [17,18,19]. During pregnancy, 5HT regulates β-cell mass [18], and it has also been shown that 5HT can modulate insulin secretion via serotonylation of GTPases [20], further supporting the association between pancreatic 5HT and insulin release. β-cells have the capacity for 5HT synthesis de novo, expressing tryptophan hydroxylase (TPH), the key enzyme in 5HT synthesis, serotonin transporter (5HTT), the main regulator of 5HT action, as well as different types of 5HT receptors [21,22]. Among 5HT receptors, signalling through the 5HT2B receptor seems to be critical for the effect of 5HT on the pancreas observed in pregnancy [18]; moreover, its activation in mouse and human β-cells enhanced glucose-stimulated insulin secretion in vitro [23] and in vivo [24]. 

In this study, we aimed to encompass the impact of both 5HT and BMP6 on glucose metabolism. The direct effect of BMP6 and 5HT on pancreatic β-cells was studied using rat insulinoma cell line INS-1 832/13 as a model in vitro. In order to study the impact of the complete absence of BMP6 on glucose and 5HT metabolism in vivo, mice with globally inactivated *Bmp6* gene (BMP6−/−) were used, with specific focus on organs related to 5HT synthesis (gut) and catabolism (liver), as well as to the regulation of the glucose metabolism (pancreas). Our in vivo studies also contribute to the more detailed characterisation of BMP6−/− mice in regard to the possible metabolic changes as compared to their wild-type (WT) littermates. 

## 2. Results

### 2.1. Exogenous BMP6 and 5HT Have Opposite Impacts on Insulin Secretion and Glucose Metabolism in Pancreatic Endocrine Cells

To explore the impact of BMP6 and 5HT on insulin secretion and their mechanism of action on glucose metabolism in pancreatic endocrine cells, we performed a static insulin secretion assay in vitro. After 24 h treatment of INS-1 cells with BMP6 (100 ng/mL) and 5HT (1 and 10 µM) in low (2.8 mM) and high (16.7 mM) glucose conditions, supernatants for insulin ELISA analysis were collected and cell lysates were homogenised in TRIzol reagent for gene expression analysis. In high glucose conditions, BMP6 treatment in the pancreatic endocrine cells significantly decreased insulin secretion (Figure 1A) and increased the expression level of *G6Pase* with no change in the expression of genes encoding other glucose metabolic enzymes (Figure 1B). In contrast, 5HT treatment in a dose-dependent manner significantly increased insulin secretion in high-glucose conditions (Figure 1C); meanwhile, in low-glucose conditions, only the 1 µM dose of the 5HT treatment significantly increased insulin secretion (Figure 1C). The lowest dose of 5HT (1 µM) significantly decreased the expression level of *G6Pase* mRNA, and a similar trend was observed with a higher concentration of 5HT (10 µM) but without statistical significance (Figure 1D).

### 2.2. Exogenous BMP6 Has a Positive Effect on Serotonin Secretion and a Negative Effect on the Expression of the 5HT2B Receptor in Pancreatic Endocrine Cells

To explore the potential impact of BMP6 on 5HT secretion and the mechanism of its action on serotonin metabolism in pancreatic endocrine cells, we conducted a static 5HT secretion assay. After 24 h treatment of INS-1 cells with BMP6 (100 ng/mL) and increasing glucose concentrations (0 mM, 8.3 mM, and 16.7 mM), supernatants were collected for 5HT ELISA analysis and cell lysates were homogenised in TRIzol reagent for the gene expression analysis. When the highest glucose concentration (16.7 mM) was added to the medium, 5HT secretion was significantly increased after the treatment with BMP6 (Figure 1E). The expression level of *5ht2b* mRNA was decreased in the presence of BMP6 with a moderate to high glucose concentration (8.3 mM and 16.7 mM) compared to the untreated control. A similar trend was observed in the expression level of *5htt* mRNA but without statistical significance (Figure 1F).

### 2.3. Effect of BMP6 Deficiency on Glucose Homeostasis in Mice

To explore the potential effect of global BMP6 protein deficiency on glucose homeostasis in mice, we performed metabolic tests (glucose tolerance test (GTT) and insulin tolerance test (ITT)) on BMP6−/− mice as the tested group and WT mice as the control. GTT and ITT were performed on 2–3 month-old BMP6−/− and WT mice after the intraperitoneal injection of glucose (2 g/kg body weight) or insulin (1 unit/kg body weight), respectively, and the blood glucose concentration was measured at pre-determined time points. In comparison to the WT mice, female BMP6−/− mice did not show signs of glucose intolerance (Figure 2A); however, they had early signs of insulin resistance at time points of 60 and 90 min (Figure 2B). In contrast, male BMP6−/− mice did not show any differences in GTT and ITT when compared to male WT mice (Figure 2C,D) but had significantly increased insulin levels in plasma, which was not the case in female BMP6−/− mice (Figure 3A). Histological changes which could suggest a basis for diabetes development, such as different shapes and boundaries of Langerhans islets, loss of β-cells, amyloid deposits, inflammation, and fibrosis [25], were not observed in the pancreas of BMP6−/− in comparison to WT mice (Figure 3B). Relative mRNA expression of glucose metabolism-related genes (*Pepck*, *G6Pase*, *Glut2*, *Gck*, and *Foxo1*) in the pancreas and liver of both female and male BMP6−/− mice did not show statistically significant differences when compared to WT mice (Figure 3C,D). However, a decreased expression of *G6Pase* and *Glut2* mRNA levels in the pancreas was observed in male BMP6−/− mice compared to WT males (Figure 3C). A similar trend of decreased expression in *G6Pase* and *Glut2* mRNA levels was observed in the female and male liver samples (Figure 3D).

### 2.4. Effect of BMP6 Deficiency on Serotonin Homeostasis in Mice 

To determine whether BMP6 protein deficiency at the global level in mice affects serotonin homeostasis, we measured the concentration of serotonin in PRP and performed gene expression analysis of serotonin-related genes in the duodenum, liver, and pancreas of female and male WT and BMP6−/− mice. In addition, we performed hematoxylin and eosin (H&E) and immunohistochemical (IHC) staining of the duodenum with 5HT and 5HTT antibodies. Morphologically, there were no significant differences between female and male BMP6−/− and WT mice in the duodenum (Appendix A). 5HT-positive cells in the duodenum of female and male WT and BMP6−/− mice were observed in the submucosa and mucosa layers with a scattered distribution (Figure 4A), while 5HTT-positive cells were primarily observed in the epithelial cells of the mucosa layer (Appendix A). The intensity of staining for 5HT and 5HTT varied, showing a more robust signal in mucosa than submucosa in the case of 5HT; however, in general, there were no differences in 5HT or 5HTT signal between BMP6−/− and WT mice (Figure 4A and Appendix A). When compared to the WT mice, male BMP6−/− mice have lowered 5HT levels in PRP but they were not statistically significant (Figure 4B). Relative mRNA expression of serotonin-related genes (*Tph1, 5ht2b, 5htt,* and *5ht2c*) in the duodenum, liver, and pancreas of female and male BMP6−/− mice did not show statistically significant differences when compared to WT mice (Figure 4C,E). However, a trend towards increased expression of *5htt* mRNA level in the duodenum and *5ht2c* mRNA level in the liver was observed in BMP6−/− mice, although without statistical significance (Figure 4C,D). In addition, the expression of *Hamp*, a gene encoding the protein hepcidin, the main regulator of iron homeostasis [26,27], was downregulated in female and male BMP6−/− mice compared to WT mice (Appendix A). Histologically, the decreased production of hepcidin led to iron accumulation within hepatocytes [27,28], which was detected morphologically on the liver slides of male BMP6−/− mice (Appendix A).

## 3. Discussion

Mice with an inactivated gene-encoding BMP6 protein were first described more than 20 years ago [11]; however, there is no detailed metabolic characterisation of this animal model in the relevant literature. In the original paper, the authors of this model described delayed ossification of the sternum [11], and a later study revealed that BMP6 is required for maintaining growth plate function [29]. Further, the importance of BMP6, together with BMP7, for proper cardiac morphogenesis during embryonal development has been described [30]. BMP6 was also shown to be one of the key regulators of hepcidin expression and iron metabolism, and BMP6−/− displayed a phenotype similar to hereditary hemochromatosis [27,31]. Recently, changes in the pancreatic tissue of BMP6−/− mice have been described, showing iron deposits in the exocrine pancreas but without any effect on the endocrine pancreatic tissue [12]. Although this study did not show diabetic alterations in BMP6−/− mice, an earlier study of our group showed the effect of BMP6 addition on plasma insulin elevation and improved glucose regulation in *ob/ob* mice [10].

In order to more thoroughly explore the possible impact of the complete absence of BMP6 on glucose metabolism in mammals, we used the global BMP6−/− mouse model. In parallel, using the same model, we searched for a possible effect of BMP6 on the peripheral serotonin system, which appears to have a significant impact on insulin secretion [18,20]. Our previously published study demonstrated the association between constitutionally elevated blood serotonin and the symptoms of type 2 diabetes, namely, increased plasma levels of glucose and insulin, as well as higher glucose concentrations measured in metabolic assays [16]. It has been shown that 5HT increases hepatic gluconeogenesis [32], contributing to the increased hepatic glucose output. Furthermore, 5HT is involved in various aspects of liver function such as fat deposition, lipogenesis, and fibrosis development [33].

As expected, our results showed significantly decreased hepcidin mRNA expression in the liver of both male and female BMP6−/− mice, which confirmed previously known results [27,31,34]. In parallel, iron accumulation was observed in the hepatocytes of BMP6−/− mice. The impact of BMP6 loss on iron metabolism was obvious already at an early adult age in 3-month-old mice. However, it seems that the systemic deletion of BMP6 does not affect 5HT and glucose metabolism directly. In this study, a statistically insignificant trend towards decreased 5HT levels in plasma was observed in male BMP6−/− mice, whereas BMP6−/− female mice did not differ in circulating 5HT levels from their WT littermates. 

It is well-known that more than 90% of peripheral 5HT is synthesised in the enterochromaffin cells in the gastrointestinal tract, from where it is released to the circulation and rapidly taken up by blood platelets [35]. Besides cells of the gut mucosa, pancreatic β-cells also have the capacity for autonomous 5HT synthesis and uptake and express 5HT receptors; however, the role of 5HT in the pancreas is primarily paracrine, regulating additionally insulin secretion [32]. From the literature, it is known that pancreatic 5HT secretion depends, among other factors, on sex [36], but the overall contribution of pancreatic 5HT to systemic circulation is negligible. Further, intrinsic 5HT in β-cells is produced primarily during pregnancy [18,37,38], when not only a rise in pancreatic 5HT secretion was observed, but also a rapid turnover of secreted 5HT [19] because of the fast cytoplasmic degradation of 5HT by monoamine oxidase [22]. In adult murine β-cells, 5HT was present at a very low level, as detected by immunostaining [37]. 

From our in vitro results, we can conclude that the addition of exogenous BMP6 increases 5HT secretion from INS-1 cells but only in conditions with a high-glucose concentration; meanwhile, in low-glucose conditions, there was no difference in 5HT secreted to the medium between BMP6-treated and control cells. Similarly, we observed a trend towards lowered 5HT values in the PRP of BMP6−/− male mice but they were not statistically significant. In contrast, in females, no difference between BMP6−/− and their WT littermates was observed, despite a trend towards an increased expression level of *5htt* mRNA in the duodenum. Although the majority of 5HT measured in platelets originates from the gut [35,39], this potential difference in expression of duodenal *5htt* obviously did not affect circulatory 5HT level. It should be considered that the expression and activity of 5HTT are differently regulated throughout tissues, not only at the transcription level but also via posttranslational mechanisms [40,41]. Besides the synthesis of 5HT in the gut, the level of 5HT measured in PRP depends also on platelet 5HTT [35], which was not specifically studied at this point. In our in vitro experiments, decreased *5ht2b* mRNA expression was observed, while the secretion of 5HT was increased upon BMP6 treatment. A similar observation was described in the study of Schraenen et al. (2010) [19], where enhanced 5HT synthesis in mouse pregnancy was accompanied by very low expression levels of *5ht2b* and *5ht1d* receptors. In our study, expression of *Tph1* mRNA remained unchanged; therefore, the increased 5HT level, measured in the cell medium, is not the consequence of increased 5HT synthesis. It is more likely that increased 5HT is related to the complex regulation of the pancreatic 5HT system, which involves the regulation of 5HT release and the activity of various receptors and transcription factors [42,43]. Decreased *5ht2b* expression could be related to the associated changes in the expression of other regulatory elements, and this will be the subject of our future studies in this field. 

Taken together, it seems that BMP6 has an impact on 5HT metabolism; however, its effect depends on the tissue observed, without systemic influence on the peripheral 5HT system. This corresponds to the results obtained in vitro, where an effect of BMP6 on 5HT secretion was not observed in basal conditions.

Interestingly, in BMP6−/− mice, we observed a trend towards a higher expression level of the mRNA-encoding 5HT2C receptor, a receptor important in metabolic regulation [44,45]. Although literature data suggest an important role of the pancreatic 5HT2C receptor in the development of type 2 diabetes [45], a trend toward increased *5ht2c* mRNA expression in the pancreas and liver observed in our mouse model was not accompanied by diabetic symptoms. In vivo metabolic tests (GTT and ITT) did not indicate symptoms of diabetes or insulin resistance in BMP6−/− mice. A potential for developing insulin resistance was observed after ITT at the 60 and 90 min time points in BMP6−/− females; however, this result does not conclusively establish these animals as insulin-resistant, as their blood glucose level at earlier time points did not differ from their WT littermates.

Regarding glucose metabolism, BMP6−/− mice did not develop any symptoms of diabetes, at least not until 3 months of age. Their blood glucose level did not differ from WT mice, and slightly higher plasma insulin concentration was observed in BMP6−/− males only. This result is consistent with the result obtained on the pancreatic β-cell line, where the addition of exogenous BMP6 decreased insulin secretion into the cell medium. At first, increased plasma insulin levels in male BMP6−/− mice (Figure 3A), without concomitant difference in basal glucose levels (Figure 2C), seems paradoxical; however, in the literature, there are suggestions that hyperinsulinemia in mice could be due to the impaired insulin clearance [46] or increased free fatty acid concentration in plasma [47]. In addition, the difference in the plasma insulin level between BMP6−/− male and female mice could be due to the effect of estrogens since enhanced insulin sensitivity was observed in premenopausal women [48]. Nevertheless, it is important to include animals from both sexes in this type of study in order to better understand the complex mechanism of endocrine regulation [49]. Further studies on this animal model should also include male and female animals in parallel [50]. In addition, future studies should also include measurements of plasma-free fatty acids [47], as well as the expression of insulin-degrading enzymes in the liver [46], which should contribute to a better understanding of BMP6’s influence on insulin metabolism in vivo.

Regarding the expression of mRNA-encoding gluconeogenic enzymes in vivo, there were no significant differences between BMP6−/− and WT mice in the pancreas and liver; however, a trend towards decreased expression level of *G6Pase* mRNA in the pancreas and liver of BMP6−/− mice was observed. This is also consistent with results in vitro, where the addition of BMP6 to the medium increased the expression of *G6Pase* mRNA in INS-1 cells. However, this is in contrast to the previous study, which suggested that BMP6 improves insulin secretion [10], as well as to the literature data [9,51,52]. This discrepancy could be explained by a difference in the animal models used, since there are, to our knowledge, no studies considering diabetes or other glucose metabolism disturbances in BMP6−/− mice. 

In addition, we measured the expression level of mRNA encoding for FOXO1, a transcriptional factor known to be highly expressed in β-cells and insulin-responsive tissues such as the liver, adipose tissue, and skeletal muscle [53,54]. Its activity has been associated with the induction of gluconeogenic enzymes such as PEPCK and G6Pase [53,55]; however, in our animal model, we did not observe changes in *Foxo1* mRNA expression between BMP6−/− and WT mice. On the other hand, faster recovery of blood glucose in the insulin tolerance test was observed in BMP6−/− females, indicating the potential for developing insulin resistance, possibly at an older age, when the impairment of β-cell function and, consequently, diabetic changes are more likely to occur [56,57]. Further studies should include measurements of the expression of different glucose transporters across the different tissues [58], which could reveal possible mechanisms underlying the discrepancy in plasma insulin levels and insulin tolerance in male and female BMP6−/− mice. It is likely that global BMP6 deficiency does not cause significant changes in glucose metabolism, especially at the early adult age (3 months), but could act indirectly. Although literature data imply that BMP inhibits gluconeogenesis in vitro [10,51], the previous study did not show any diabetic phenotype even in older BMP6−/− mice [12], suggesting that the potential effect of BM6 on glucose metabolism in vivo develops slowly and is too small to have a significant impact during the lifespan of mice.

This study also has potential limitations. For one, there is large variability in the results of gene expression, which leads to statistically insignificant results and, therefore, we have to be cautious about their interpretation. Including more biological replicates could help in lowering standard errors of measurement. Our in vitro model included only one cell line, INS-1 832/13, as the most appropriate cell line for this type of investigation [59,60]. In future studies, it would be useful to explore the primary culture of pancreatic β-cells in addition to the already used INS-1 cell line, which would potentially present a more reliable model in vitro. Besides limitations in vitro, we should mention that there is the potential compensatory effect of other BMPs (for example, BMP7 or BMP9; see Introduction) in the BMP6-deficient mice, and this should be included in future studies of this animal model.

## 4. Materials and Methods

### 4.1. Cell Culture 

The rat insulinoma cell line INS-1 832/13 (Sigma–Aldrich, St. Louis, MO, USA) was cultured in complete RPMI 1640 medium with L-glutamine and sodium bicarbonate (Sigma Aldrich, St. Louis, MO, USA) supplemented with 10% dialysed fetal bovine serum (FBS), 10 mM HEPES, 2 mM glutamine, 0.05 mM β-mercaptoethanol, 1 mM sodium pyruvate, 10 mg/mL streptomycin, 25 µg/mL amphotericin B, and 10,000 U/mL penicillin [61]. Cells were cultured at 37 °C in a humidified atmosphere containing 95% air and 5% CO_2_. For static insulin and serotonin secretion assays, cells were cultured in a 48-well plate at a density of 1 × 10^5^ cells/mL/well in RPMI 1640 media described above.

### 4.2. Static Insulin and Static Serotonin Secretion Assay

For the insulin secretion assay, cultured INS-1 cells were treated with BMP6 (100 ng/mL) or 5HT (1 and 10 µM) for 24 h. After the treatment, cells were washed and preincubated in Krebs–Ringer buffer (KRB) without glucose for 2 h at 37 °C. Following preincubation, the cells were divided into two groups and treated with KRB buffer with the addition of a low (2.8 mM) or high (16.7 mM) concentration of glucose for a further 2 h at 37 °C. For the serotonin secretion assay, INS-1 cells were cultured in two 48-well plates and incubated with a complete RPMI 1640 containing 1% dialysed FBS for 24 h. The serum-starved cells in both 48-well plates were divided into three groups and stimulated with complete RPMI 1640 medium containing 1% dialysed FBS without added glucose (control), or with the addition of 8.3 mM or 16.7 mM glucose concentration for 24 h. At the same time, one plate was treated with BMP6 (100 ng/mL) while another plate represented an untreated control. 

The supernatants from both assays were collected from each well after the treatment and stored at −20 °C for further insulin and serotonin measurements using a commercial ELISA insulin kit (rat insulin kit: 10-1250 Mercodia, Uppsala, Sweden; mouse insulin kit: ab285341 Abcam, Cambridge, UK) and ELISA serotonin kit (BA E-5900, LDN, Nordhorn, Germany) according to the manufacturer’s instructions. After the collection of supernatants, cells were lysed, homogenised in TRIzol reagent (Invitrogen, Waltham, MA, USA), and stored at −80 °C for RNA isolation and further gene expression analysis.

### 4.3. Animals and Animal Care

Wild type (WT) and BMP6 knockout (BMP6−/−) C57BL/6J male and female mice (Charles River, MA, USA) were housed under a standard 12 h light/dark cycle in a climate-controlled environment. Standard chow diet and water were given ad libitum. Genotyping was performed using DNA isolated from mouse tail tissue using real-time PCR with complete DreamTAQ Hot Start Green Master Mix (Thermo Fisher, Waltham, MA, USA). The primers used for genotyping were one upstream BMP6 primer (5′-TCCATGATCCCTCTAACTCG-3′) and two downstream BMP6 primers (5′-GCTGATGACAGCAGCCATTG-3′ for the WT allele and 5′-TTACACACAGCATGCTCACC-3′ for the mutant allele). 

### 4.4. Glucose Tolerance Test (GTT) and Insulin Tolerance Test (ITT)

GTT and ITT were performed in 2–3-month-old female and male WT and BMP6−/− mice (n = 17–20 females and 12 males per group for GTT and n = 14–15 females and males per group for ITT). For the GTT, after an overnight fast, the female and male WT and BMP6−/− mice were injected intraperitoneally with D-glucose solution (Sigma Aldrich, St. Louis, MO, USA) (1 g of glucose/kg body weight). For the ITT, after a 6-h fast, the female and male WT and BMP6−/− mice were injected intraperitoneally with Humulin R insulin solution (Lilly France S.A.S., Fegersheim, France) (1 unit/kg body weight). Blood glucose concentrations were directly measured from the tail vein using a glucometer (AccuCheck Performa, Roche, Basel, Switzerland) before the injection at basal (0 min) and after the injection at serial time points 15, 30, 60, 90 for ITT, as well as for an additional time point of 120 min for GTT [62]. The ethical principles of the study are aligned with ARRIVE guidelines and European Directive 2010/63/EU for animal experiments. 

Upon completion of metabolic tests, WT and BMP6−/− mice were sacrificed through a terminal cardiac puncture procedure, during which blood samples were collected in Vacuette^®^ tubes (Greiner Bio-One, Kremsmünster, Austria) with the K3EDTA additive for further ELISA analyses. The terminal cardiac puncture was performed under deep general anaesthesia of mice with a combination of ketamine (80 mg/kg body weight) and xylazine (10 mg/kg body weight) injected intraperitoneally. Blood samples were centrifuged according to the manufacturer’s recommended ELISA protocols and plasma or platelet-rich plasma (PRP) samples were collected. Plasma was used for the insulin ELISA analysis while PRP was used for serotonin ELISA analyses. After the blood sample collection, samples of pancreas, liver, and duodenum tissues were taken. Tissue samples for gene expression analyses were homogenised in TRIzol solution and stored at −80 °C, while tissue samples for histological staining and immunohistochemistry were fixed in 10% formalin solution (Sigma–Aldrich, St. Louis, MO, USA) for 24 h at room temperature. After fixation, tissues were washed with distilled water, dehydrated in increasing concentrations of ethanol (70% to 100%), and embedded in paraffin wax. Paraffin-embedded tissues were sliced into 5 µm sections. 

### 4.5. Gene Expression Analyses

From INS-1 cells and homogenised tissue samples, total RNA was isolated by using a TRIzol reagent according to the manufacturer’s protocol. The purity and the concentration of RNA were determined by measuring the absorbance at 260 nm with the Biophotometer (Eppendorf, Hamburg, Germany). cDNA was synthesised using 1 µg of total RNA, oligo (dT) primers, and the High-Capacity cDNA Reverse Transcription kit (Thermo Fisher, Waltham, MA, USA) using the GeneAmp Thermal Cycler 2400 (Perkin–Elmer, Waltham, MA, USA). Quantitative real-time PCR was conducted using cDNA, forward and reverse primers, and the LightCycler FastStart DNA Master Syber Green kit (Roche, Basel, Switzerland) using the LightCycler 1.5 (Roche, Basel, Switzerland) as described [63,64]. The specific rat and mouse primers used in gene expression analysis are given in Appendix A.

### 4.6. Histological Analysis and Immunohistochemistry

Histological slides of the pancreas, duodenum, and liver of female and male BMP6−/− and WT mice were stained with hematoxylin and eosin (H&E) using standard protocols [65]. For the preparation of slides, three mice from each group (n = 3) were used, with three to four tissue sections per slide. Immunohistochemistry (IHC) was conducted with rabbit-specific HRP/DAB IHC detection kit–micro-polymer (Abcam, Cambridge, UK) and the following primary antibodies: anti-5HT antibody (rabbit, dilution factor 1:20,000 in PBS/0.3% Triton X-100, Immunostar, Hudson, WI, USA), and anti-5HTT antibody (rabbit, dilution factor 1:500, Abcam, Cambridge, UK). After the H&E and IHC staining, images were obtained and analysed using an Olympus BX53 microscope (Hamburg, Germany) with a DP27 camera at 20× and 40× magnification.

All statistical analyses were performed using GraphPad Prism 8.4.3 software. The level of statistical significance between the means of the two populations was determined using unpaired Student’s two-tailed *t*-test, multiple *t*-tests with post-hoc Holm–Sidak, Welch’s *t*-test, or a Mann–Whitney U-test depending on the data distribution. The comparative CT method (ΔΔCT) was used to determine the amount of target gene, normalised to a “housekeeping” reference gene (β-actin for INS-1 cell supernatants, GAPDH for homogenised tissues samples) and relative to a calibrator. Relative mRNA expression was presented as a fold change + SEM. All the other data were presented as mean ± SEM. The results were considered significant when *p* < 0.05.

## 5. Conclusions

BMP6 is associated with changes in iron metabolism, only marginally affects 5HT metabolism, and has almost no influence on glucose metabolism. While systemic deletion of BMP6 significantly affects the liver expression of hepcidin, there are no effects of this deletion on the expression of genes involved in 5HT or glucose metabolism regulation in any of the organs explored (pancreas, liver, duodenum). However, slight changes in the systemically circulating 5HT level and insulin tolerance test in BMP6−/− mice were observed, suggesting the possibility that BMP6, via subtle changes in the peripheral 5HT system, could gradually affect the glucose metabolism, becoming more enhanced with ageing.

## Figures and Tables

**Figure 1 ijms-25-07842-f001:**
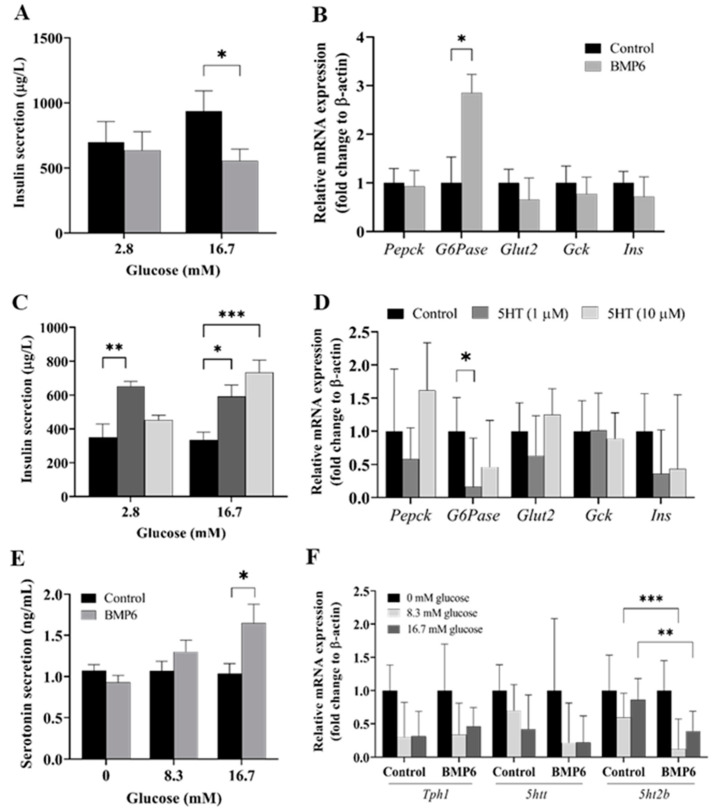
Static insulin and serotonin secretion assays in INS 832/13 rat insulinoma cells: (**A**) Supernatant insulin secretion in low (2.8 mM) and high (16.7 mM) glucose conditions following treatment with BMP6 (100 ng/mL) (n = 8) was measured with ELISA, while (**B**) relative mRNA expression of glucose metabolism-related genes was measured in cell lysate with RT-qPCR (n = 12). (**C**) Supernatant insulin secretion in low (2.8 mM) and high (16.7 mM) glucose conditions following treatment with 5HT (1 µM and 10 µM) (n = 6) was measured with ELISA, while (**D**) relative mRNA expression of glucose metabolism-related genes was measured in cell lysate with RT-qPCR (n = 4). (**E**) Supernatant serotonin secretion with increasing glucose concentrations (0 mM, 8.3 mM and 16.7 mM) following the treatment with BMP6 (100 ng/mL) was measured with ELISA (n = 6), while (**F**) relative mRNA expression of serotonin metabolism-related genes was measured in cell lysates with RT-qPCR (n = 8). Results of relative mRNA expression are reported as fold change + SEM, while ELISA results are reported as mean + SEM. * *p* < 0.05, ** *p* < 0.01, *** *p* < 0.001.

**Figure 2 ijms-25-07842-f002:**
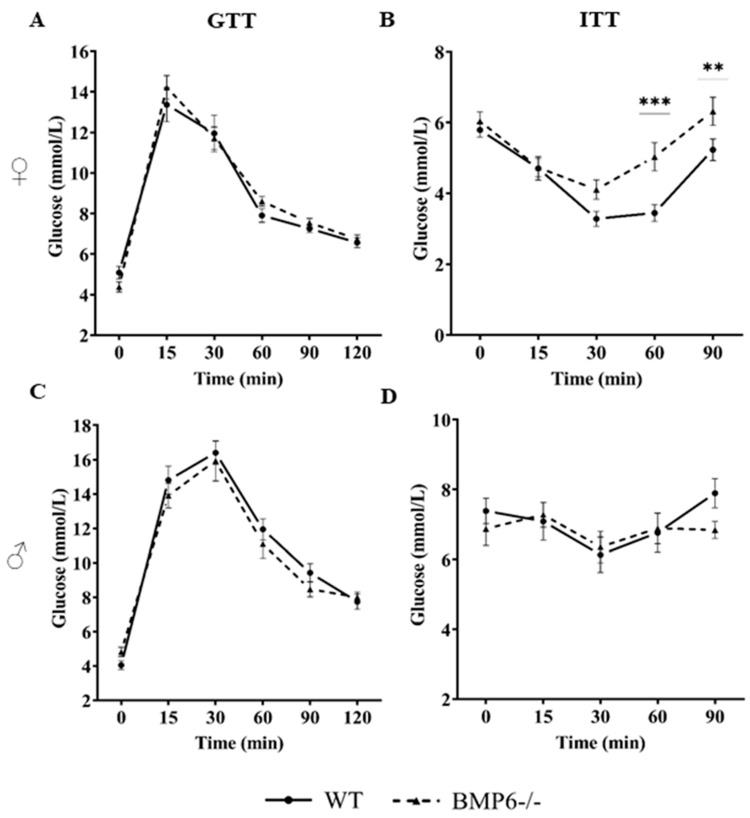
Metabolic tests in BMP6−/− and WT mice. GTT in (**A**) female and (**C**) male BMP6−/− mice compared to WT mice (n = 17–20 females and n = 12 males). ITT in (**B**) female and (**D**) male BMP6−/− mice compared to WT mice (n = 14–15). Blood glucose concentration was measured in both metabolic tests. Results are reported as mean ± SEM. ** *p* < 0.01 *** *p* < 0.001.

**Figure 3 ijms-25-07842-f003:**
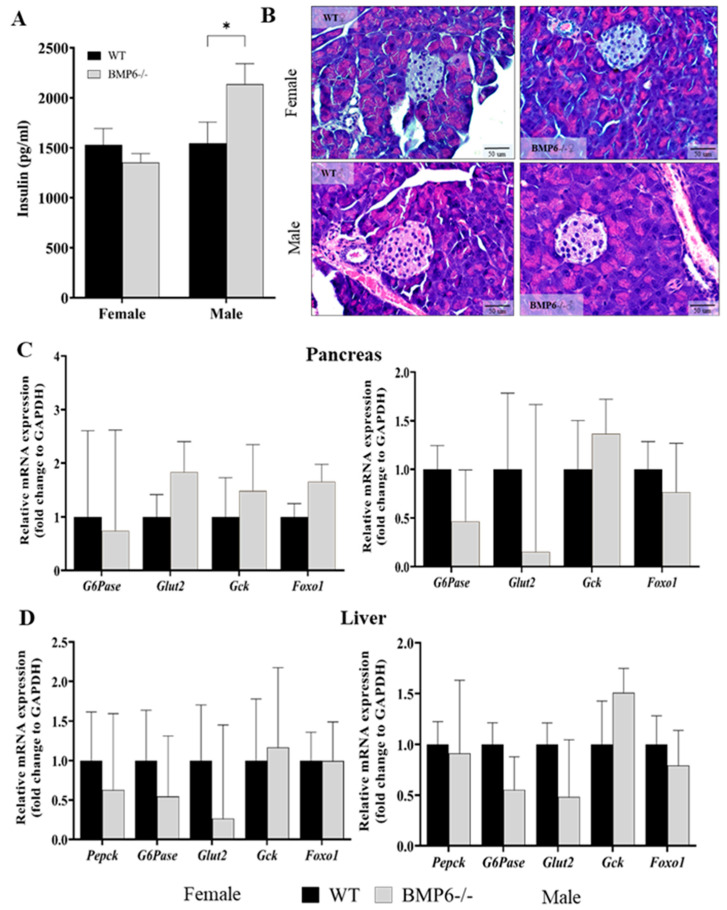
Effect of BMP6 deficiency on glucose homeostasis in mice: (**A**) Plasma insulin level in female and male BMP6−/− mice compared to WT mice (n = 8). (**B**) H&E staining of the Langerhans islets of the pancreas from female and male WT and BMP6−/− mice, magnification 40×, scale bar = 50 µm (n = 3). (**C**,**D**) Relative mRNA expression of glucose metabolism-related genes was measured with RT-qPCR in the (**C**) pancreas and (**D**) liver of female and male WT and BMP6−/− mice (n = 7–11). Results are reported as fold change + SEM for relative mRNA expression and mean + SEM for plasma insulin levels. * *p* < 0.05.

**Figure 4 ijms-25-07842-f004:**
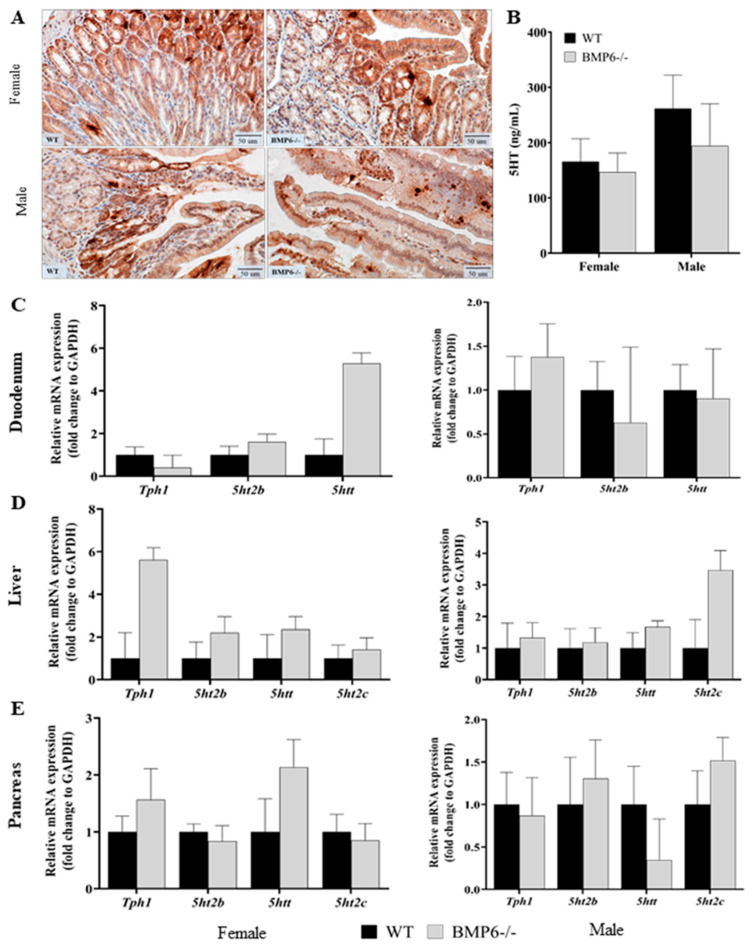
Effect of BMP6 deficiency on serotonin (5HT) metabolism in mice: (**A**) Immunohistochemical analysis of 5HT (a distinct brown/red colour) in the duodenum of female and male WT and BMP6−/− mice, magnification 40×, scale bar = 50 µm (n = 3). (**B**) PRP serotonin level in female and male BMP6−/− compared to WT mice (n = 7–12). (**C**,**E**) Relative mRNA expression of serotonin-related genes was measured with RT-qPCR in the (**C**) duodenum, (**D**) liver, and (**E**) pancreas of female and male WT and BMP6 mice (n = 7–11). Results are reported as fold change + SEM for relative mRNA expression and mean + SEM for plasma serotonin levels.

## Data Availability

The raw data supporting the conclusions of this article will be made available by the authors upon request.

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
