# Peer review of "The Influence of BMP6 on Serotonin and Glucose Metabolism"

_ijms, 2024, doi:10.3390/ijms25147842_

Round 1

Reviewer 1 Report

Comments and Suggestions for Authors

The submitted manuscript is an investigation of the effect of BMP6 and serotonin on glucose metabolism. Experiments were performed in vitro and in vivo, using biochemical and molecular techniques. The authors conclude that BMP6 has a minor effect on glucose metabolism.

Overall, the results presented are non-significant. However, negative results are also important in science.

Nevertheless, the paper has some points to take into consideration. 

1. why didn't the authors make a curve to determine the appropriate concentration to use in the BMP6 experiments? or explain how the concentration used was decided?

2. The authors should discuss why, according to Figure 1E and 1F, the higher the concentration of serotonin, the lower the expression of its receptor.

3. The difference found between male and female should be discussed in more detail, since it was one of their main results.

4. The authors state in their conclusion that BMP6 has a direct effect on iron metabolism. However, this statement seems to me to be exaggerated because they only found one gene differentially expressed with this process. I suggest being more cautious with their conclusion or extending the study to other genes related to this process.

5. Some of the experiments were performed with a small n, for example 3. The authors should justify this point and mention if repetitions of the experiments were performed. 

6. Among the minor points are to include more information on the reagents and antibodies used, and to include the limitations of the work.

Author Response

The submitted manuscript is an investigation of the effect of BMP6 and serotonin on glucose metabolism. Experiments were performed in vitro and in vivo, using biochemical and molecular techniques. The authors conclude that BMP6 has a minor effect on glucose metabolism.

Overall, the results presented are non-significant. However, negative results are also important in science.

Nevertheless, the paper has some points to take into consideration. 

  1. why didn't the authors make a curve to determine the appropriate concentration to use in the BMP6 experiments? or explain how the concentration used was decided?

The BMP6 concentration used in described experiments was the same as used in our previous study (Pauk et al. Acta Diabetologica 2018) when exploring the effect of BMP6 in vitro, and the same concentration (100 ng/mL) of BMP7 was used on human islet cells in vitro in the study of Klein et al. (Diabetes, 2015). Therefore, we decided to use the concentration of 100 ng/mL which already proved to be effective in similar models in vitro.

  1. The authors should discuss why, according to Figure 1E and 1F, the higher the concentration of serotonin, the lower the expression of its receptor.

We are thankful to the reviewer for this observation. Indeed, in high glucose conditions (16.7 mM), secretion of 5HT from INS-1 cells increased, whereas expression of 5ht2b mRNA was significantly decreased. This at first seems paradoxically, especially when considering the role of 5HT2B receptor in regulation of serotonergic neuron activity. However, regulation of serotonin activity in pancreas is complex,  controlled by several factors and it is probably not regulated only at transcriptional, but also at post-translational level. In study of Schraenen et al. (2010), expression of 5ht2b mRNA was at very low levels, not parallel with strong induction of serotonin biosynthesis. This is similar to our results shown in Figure 1F; however, in our study, we did not observe increase in Tph2 mRNA expression. It is possible that increased levels of 5HT measured in the cell medium are a consequence of enhanced 5HT release from the cells. Decreased expression of 5ht2b could be related to the associated changes in expression of other regulatory elements, and this will be the subject of future studies in this field.

This has been added to the Discussion section (lines 251-261) and we believe that this explanation would contribute to the overall discussion of the results.

  1. The difference found between male and female should be discussed in more detail, since it was one of their main results.

Thank you for this suggestion. We have additionally discussed the difference between male and female mice in the Discussion section (lines 280-293).

  1. The authors state in their conclusion that BMP6 has a direct effect on iron metabolism. However, this statement seems to me to be exaggerated because they only found one gene differentially expressed with this process. I suggest being more cautious with their conclusion or extending the study to other genes related to this process.

We are thankful to the reviewer for this suggestion. Yes, this conclusion really sounds exaggerated and should be reformulated. In the revised version of the manuscript, this sentence is rephrased (line 430).

  1. Some of the experiments were performed with a small n, for example 3. The authors should justify this point and mention if repetitions of the experiments were performed. 

The experiments reviewer is referring to are immunohistochemical stainings depicted in Figures 3B and 4A. For the preparation of H&E and IHC slides, three mice from each group were used (n=3), with three to four tissue sections per slide. These experiments did not include quantitative analyses, otherwise, more samples would have been included. Immunohistochemical analyses in this study were performed at qualitative level and only a representative picture for each staining has been shown.

In order to clarify this issue, we have added a sentence in the Materials and Methods section in the revised version of the manuscript (lines 412-413).

  1. Among the minor points are to include more information on the reagents and antibodies used, and to include the limitations of the work.

We have added details about ELISA kits used in Materials and Methods section (line 357) and included the paragraph at the end of Discussion section (lines 321-331) about the limitations of the work.

Reviewer 2 Report

Comments and Suggestions for Authors

In this study, the authors investigated the impact of both 5HT and BMP6 on glucose metabolism. The authors concluded that BMP6 does not directly influence glucose metabolism, but there is a possibility that its deletion causes slowly developing changes in glucose and serotonin metabolism, which would become more expressed with aging.

Comments

The reviewer has some concerns as follows:

1.     One of the major concerns is the correctness of data presentation. The variations of some data from in vitro and in vivo experiments are very large under mean±SEM situation, leading to no significant difference between control and tested groups. Moreover, in the Methods, it described that GTT and ITT were …female and male WT and BMP6-/- mice (n=12-14); however, in the Figure 2, it described that GTT in… ITT in (B) female and (D) male BMP6-/- mice compared to WT mice (n=12-20); what are the correct n values?

2.     The authors have published their study as a preprint version on SSRN (https://papers.ssrn.com/sol3/papers.cfm?abstract_id=4712934). In the Figure 1 of this preprint version, the n numbers in (D) and (F) are 6 and 12, respectively. However, the same data in the Figure 1 of the present manuscript, the n numbers in (D) and (F) are 4 and 8, respectively. The reviewer’s concern is why the n numbers are different, but the data shown in figure are same. Moreover, the standard errors (SE) or deviations (SD) of data are confusing in these two versions.

3.     In Figure 2B, there is the changes in insulin tolerance test (ITT) in female BMP6-/- mice, but why the blood insulin levels increased in male BMP6-/- mice? There seems an inconsistent gender difference.

4.     Overall, there is unreliability in the data presentation. The presented results cannot support the conclusions.

Author Response

In this study, the authors investigated the impact of both 5HT and BMP6 on glucose metabolism. The authors concluded that BMP6 does not directly influence glucose metabolism, but there is a possibility that its deletion causes slowly developing changes in glucose and serotonin metabolism, which would become more expressed with aging.

Comments

The reviewer has some concerns as follows:

  1. One of the major concerns is the correctness of data presentation. The variations of some data from in vitro and in vivo experiments are very large under mean±SEM situation, leading to no significant difference between control and tested groups. Moreover, in the Methods, it described that GTT and ITT were …female and male WT and BMP6-/- mice (n=12-14); however, in the Figure 2, it described that GTT in… ITT in (B) female and (D) male BMP6-/- mice compared to WT mice (n=12-20); what are the correct n values?

We thank to the reviewer for this remark and agree that the number of animals used in experiments sounds confusing when comparing description in Materials and Methods and Figure 2. The correct n values are:

Figure 2A n (WT) = 17, n (BMP6) = 20

Figure 2B n = 15 (both groups)

Figure 2C n = 12 (both groups)

Figure 2D n ( WT) = 15, n (BMP6) = 14

The variability in n values between experiments is due to the exclusion of outlier values in experiments, as well as exclusion of mice younger than 7 weeks.

In the revised version of the manuscript, n values are more precisely defined in both Materials and Methods section (lines 373-374) and in the Figure 2 description (lines 148-149).

  1. The authors have published their study as a preprint version on SSRN (https://papers.ssrn.com/sol3/papers.cfm?abstract_id=4712934). In the Figure 1 of this preprint version, the n numbers in (D) and (F) are 6 and 12, respectively. However, the same data in the Figure 1 of the present manuscript, the n numbers in (D) and (F) are 4 and 8, respectively. The reviewer’s concern is why the n numbers are different, but the data shown in figure are same. Moreover, the standard errors (SE) or deviations (SD) of data are confusing in these two versions.

After a careful examination of the first version of the article, we have found some errors in figure description, which were corrected in the present version of the article. This is the reason for different n values. Further, we decided to use only standard errors (SE) throughout the entire article, instead using SE and SD, which may be confusing.

  1. In Figure 2B, there is the changes in insulin tolerance test (ITT) in female BMP6-/- mice, but why the blood insulin levels increased in male BMP6-/- mice? There seems an inconsistent gender difference.

We agree that difference in ITT and blood insulin levels seems to be inconsistent. In the revised version of the manuscript, we discussed the observed difference in blood insulin levels between BMP6-/- and WT male mice. On the other hand, female BMP6-/- mice did not show differences in plasma insulin levels when compared to the WT mice; however, in the insulin tolerance test, BMP6-/- females recovered plasma glucose levels faster than WT mice. In fact, here we did not explore tissue expression of different glucose transporters, insulin-dependent and insulin-independent, which could be responsible for these differences. We believe that future studies which should explore expression of several glucose transporters across the different tissues could help reveal the mechanisms underlying this, at first paradoxical observation. This has now been added to the Discussion section (lines 280-293).

  1. Overall, there is unreliability in the data presentation. The presented results cannot support the conclusions.

We tried to answer adequately comments from all reviewers and have added more text in the Discussion. We believe that this could improve the overall quality of the presented results and their interpretation.

Reviewer 3 Report

Comments and Suggestions for Authors

This study investigates the potential interactions between bone morphogenetic protein 6 (BMP6), serotonin (5-hydroxytryptamine, 5HT), and glucose metabolism regulation through in vitro and in vivo experiments. Using the INS-1 832/13 rat insulinoma cell line, the researchers found that BMP6 and 5HT have opposite effects on insulin secretion from pancreatic β-cells. In vivo studies with BMP6-deficient mice (BMP6-/-) revealed that although BMP6 deficiency does not induce diabetic changes, there is a mild difference in insulin tolerance compared to wild-type mice. Additionally, BMP6 impacts the 5HT system in a tissue-specific manner without affecting systemic peripheral 5HT metabolism. Overall, this study added some key references to the field even though the phenotypes of the BMP6-deficient mice are very mild.

Here are some points that need to be addressed to improve the clarity and robustness of the study:

  1. Figure 1A and 1C:
    • Observation: There is significant variability in insulin secretion in the control group under the same conditions, making it difficult to compare with the experimental group.
    • Suggestion: Please investigate and provide an explanation for the observed variability. This could be due to differences in experimental conditions, cell confluence, or batch effects. Consider including additional replicates or normalizing the data to account for this variability to improve the comparability of the results.
  2. Figure 1C:
    • Legend Inconsistency: The figure lacks legends for the colors used, which are inconsistent with those in Figure 1D.
    • Recommendation: Add clear legends for the color traces in Figure 1C to ensure consistency and improve interpretability. This will help readers accurately understand the data presented.
  3. Figure 3A:
    • Observation: There is a significant increase in plasma insulin levels in male BMP6-deficient mice, yet the basal plasma glucose levels are similar between WT and BMP6-deficient mice (Figure 2).
    • Explanation Needed: Discuss the potential mechanisms underlying the increased insulin levels without a corresponding change in basal glucose levels. This could involve altered insulin sensitivity or compensatory mechanisms in BMP6-deficient mice. Further experiments or literature references could help elucidate this observation.
  4. Figure 3B:
    • Observation: The islet size in female mice appears significantly larger than in male mice.
    • Clarification Needed: Verify if this observation holds across all samples or if it might be due to scaling errors or islet cell degeneration in male mice. Provide additional data or images to support the consistency of this finding and ensure the accuracy of the scale bars used in the figures.
  5. Compensatory Effects:
    • Observation: The phenotype of BMP6-deficient mice is relatively mild regarding insulin and serotonin secretion.
    • Recommendation: Investigate the expression of other BMPs (such as BMP7 and BMP9) in BMP6-deficient mice as suggested in the introduction. This will help elucidate potential compensatory mechanisms and provide a clearer understanding of the observed phenotypes. Including this data will strengthen the conclusions of your study and provide insight into the redundancy and interactions within the BMP signaling pathway.

Addressing these points will significantly enhance the manuscript’s clarity, rigor, and overall impact. Ensuring accurate interpretation of results and robust experimental validation will strengthen your conclusions and provide a clearer understanding of the mechanisms at play.

Author Response

This study investigates the potential interactions between bone morphogenetic protein 6 (BMP6), serotonin (5-hydroxytryptamine, 5HT), and glucose metabolism regulation through in vitro and in vivo experiments. Using the INS-1 832/13 rat insulinoma cell line, the researchers found that BMP6 and 5HT have opposite effects on insulin secretion from pancreatic β-cells. In vivo studies with BMP6-deficient mice (BMP6-/-) revealed that although BMP6 deficiency does not induce diabetic changes, there is a mild difference in insulin tolerance compared to wild-type mice. Additionally, BMP6 impacts the 5HT system in a tissue-specific manner without affecting systemic peripheral 5HT metabolism. Overall, this study added some key references to the field even though the phenotypes of the BMP6-deficient mice are very mild.

Here are some points that need to be addressed to improve the clarity and robustness of the study:

1. Figure 1A and 1C:

  • Observation: There is significant variability in insulin secretion in the control group under the same conditions, making it difficult to compare with the experimental group.
  • Suggestion: Please investigate and provide an explanation for the observed variability. This could be due to differences in experimental conditions, cell confluence, or batch effects. Consider including additional replicates or normalizing the data to account for this variability to improve the comparability of the results.

We thank to the reviewer for this observation. The difference in insulin concentration under conditions of low (2.8 mM) and high (16.7 mM) glucose condition shown in figures 1A and 1C could be cumulative effect influenced by slightly different experimental conditions. Although the initial concentration of the cells and other experimental conditions were the same, in the experiment with 5HT treatment we have used dialyzed FBS, which has a reduced content of amino acids, hormones, and cytokines and that could potentially affect cell growth during the experiment.

2. Figure 1C:

  • Legend Inconsistency: The figure lacks legends for the colors used, which are inconsistent with those in Figure 1D.
  • Recommendation: Add clear legends for the color traces in Figure 1C to ensure consistency and improve interpretability. This will help readers accurately understand the data presented.

We thank to the reviewer for this remark. Colors in Figure 1C have been corrected to match the legend shown under Figure 1D.

3. Figure 3A:

  • Observation: There is a significant increase in plasma insulin levels in male BMP6-deficient mice, yet the basal plasma glucose levels are similar between WT and BMP6-deficient mice (Figure 2).
  • Explanation Needed: Discuss the potential mechanisms underlying the increased insulin levels without a corresponding change in basal glucose levels. This could involve altered insulin sensitivity or compensatory mechanisms in BMP6-deficient mice. Further experiments or literature references could help elucidate this observation.

We thank to the reviewer for this observation. Indeed, the difference in plasma insulin between BMP6-deficient and wild-type male mice, without a corresponding difference in basal glucose levels at first seems to be paradoxical. Further, male mice did not show any differences in glucose or insulin-tolerance tests performed, which implicate that the difference in basal insulin levels was not accompanied with differences in glucose tolerance under increased glucose load. However, there are literature references which suggest that hyperinsulinemia can appear due to the impaired insulin clearance in liver (Marmentini et al., 2021). There is also influence of free fatty acids which, when increased, stimulate insulin release (Shanik et al., 2008). Further studies on BMP6 knockout mouse model should include measurement of free fatty acids, as well as expression of insulin-degrading enzyme in the liver. This has now been included in the Discussion section (lines 280-293).

4. Figure 3B:

  • Observation: The islet size in female mice appears significantly larger than in male mice.
  • Clarification Needed: Verify if this observation holds across all samples or if it might be due to scaling errors or islet cell degeneration in male mice. Provide additional data or images to support the consistency of this finding and ensure the accuracy of the scale bars used in the figures.

All images have been taken under the same magnification. After examination of all the slides and taken pictures of Langerhans islet, we could observe that the different sizes of Langerhans islets are present in both female and male mice of both genotypes. For the paper, the most representative picture have been chosen. To avoid confusion, we have replaced Figure 3B with another image, with more uniform Langerhans islets presented. Generally, we did not observe differences in the structure of Langerhans island between male and female mice.

5. Compensatory Effects:

  • Observation: The phenotype of BMP6-deficient mice is relatively mild regarding insulin and serotonin secretion.
  • Recommendation: Investigate the expression of other BMPs (such as BMP7 and BMP9) in BMP6-deficient mice as suggested in the introduction. This will help elucidate potential compensatory mechanisms and provide a clearer understanding of the observed phenotypes. Including this data will strengthen the conclusions of your study and provide insight into the redundancy and interactions within the BMP signaling pathway.

 We appreciate reviewer's suggestion and agree that investigation of other BMPs in BMP6-deficient mice would contribute significantly to the revealing the mechanisms underlying subtle changes in metabolism. This was included as a limitation of the study at the end of the Discussion section (lines 321-331) and will be investigated in our future studies on this animal model.

Round 2

Reviewer 2 Report

Comments and Suggestions for Authors

Although the authors tried to explain the reviewer’s concerns, the authors' explanation of the n value is not acceptable.

The difference of n values between the previous version [GTT and ITT were performed…female and male WT and BMP6-/- mice (n=12-14)] and the revised version [n=17-20 females and 12 males per group for GTT and n=14-15 females and males per group for ITT] is a bit big. What are the correct n values?

The preprint version of the presented study is a publicly published paper, and the figures in it are the same as the figures in the present manuscript but have different n values. This has shown that the data is unrealistic. The statistical errors presented as SE or SD have nothing to do with the n values.

Therefore, there is unreliability in the data presentation. The presented results cannot support the conclusions.